# Oestrogen Detoxification Ability of White Rot Fungus *Trametes hirsuta* LE-BIN 072: Exoproteome and Transformation Product Profiling

**DOI:** 10.3390/jof10110795

**Published:** 2024-11-15

**Authors:** Olga S. Savinova, Tatiana S. Savinova, Tatyana V. Fedorova

**Affiliations:** Bach Institute of Biochemistry, Federal Research Center, Fundamentals of Biotechnology, Russian Academy of Sciences, 119071 Moscow, Russia; savinovatatiana@yandex.ru (T.S.S.); fedorova_tv@mail.ru (T.V.F.)

**Keywords:** *Trametes hirsuta*, oestrogens, oestrone, 17β-oestradiol, biotransformation, endocrine-disrupting chemicals, exoproteome

## Abstract

White rot fungi, especially representatives of the genus *Trametes* spp. (Polyporaceae), are effective destructors of various xenobiotics, including oestrogens (phenol-like steroids), which are now widespread in the environment and pose a serious threat to the health of humans, animals and aquatic organisms. In this work, the ability of the white rot fungus *Trametes hirsuta* LE-BIN 072 to transform oestrone (E1) and 17β-oestradiol (E2), the main endocrine disruptors, was shown. More than 90% of the initial E1 and E2 were removed by the fungus during the first 24 h of transformation. The transformation process proceeded predominantly in the direction of the initial substrates’ detoxification, with the radical oxidative coupling of E1 and E2 as well as their metabolites and the formation of less toxic dimers in various combinations. A number of minor metabolites, in particular, less toxic estriol (E3), were identified by HPLC-MS. The formation of E1 from E2 and vice versa were shown. The exoproteome of the white rot fungus during the transformation of oestrogens was studied in detail for the first time. The contribution of ligninolytic peroxidases (MnP5, MnP7 and VP2) to the process of the extracellular detoxification of oestrogens and their possible metabolites is highlighted. Thus, the studied strain appears to be a promising mycodetoxicant of phenol-like steroids in aquatic environments.

## 1. Introduction

The development of safe methods for the utilisation and neutralisation of various xenobiotics in order to reduce environmental pollution is an urgent task of modern biotechnology. Particular attention is paid to the problem of contamination with biologically active substances belonging to the class of oestrogens [1]. Oestrogens belong to the group of typical endocrine disrupting chemicals (EDCs). Upon getting into the organism even at low concentrations (ng/L), they can cause various physiological disorders such as the inhibition of reproduction (by disrupting the anabolism and catabolism of endogenous hormones) and have carcinogenic effects [2,3,4,5,6,7,8].

Oestrone (E1, 3-hydroxyestra-1,3,5(10)-triene-17-one) and oestradiol (E2, 17β-oestradiol, 3,17β-dihydroxyestra-1,3,5(10)-triene) are naturally occurring oestrogens regularly excreted by humans and animals and used as pharmaceutical preparations for humans and agriculture. Their structures are shown in Figure 1.

The appearance of these compounds in the environment is mainly due to their anthropogenic origin (emissions from pharmaceutical enterprises, use in agriculture, through human and animal excretions (urine and faeces), etc.) [9]. One of the causes of global soil and water pollution with E1 and E2 is the widespread use of animal manure as a fertiliser [10]. The concentrations of these compounds in the environment today reach alarming levels that threaten the natural balance of ecosystems and may be detrimental to human health [11]. Therefore, the need to remove such biologically active compounds from the environment is very acute.

The microbiological transformation of oestrogens has advantages over chemical treatment and can be considered a promising method for their neutralisation in accordance with the principles of green chemistry. A number of works have been published where the transformation of oestrogens was carried out using various microorganisms, in particular, filamentous fungi [9]. It is believed that microorganisms can degrade or biotransform steroids by three major pathways. The first pathway is associated with growth (metabolic), i.e., oestrogens are used by microorganisms as the sole source of carbon and energy. The second is non-growth related, in which case, microorganisms grow using other sources of carbon and energy found in the environment and produce enzymes that catalyse the biotransformation of oestrogen into various products. The third pathway is the conversion of steroids into metabolites without decomposing them (detoxification) [9]. At the same time, the data presented in the literature indicate that different fungi can use unique mechanisms of the biotransformation/detoxification of steroids [12], which, obviously, may be due to differences in the compositions of their enzymatic complexes.

Researchers have paid special attention to white rot fungi. These organisms, capable of degrading such a complex biopolymer as lignin and having great potential for detoxifying various biologically active pollutants, are promising in relation to the neutralisation of oestrogens (phenol-like steroids) [13,14,15,16]. It is believed that the main enzymes responsible for the oestrogen modification by white rot fungi are peroxidases (PODs) and phenol oxidases such as laccase [11,13,17,18]. However, dehydrogenase (aryl-alcohol dehydrogenase (EC 1.1.1.91)) [19], glucose dehydrogenase (GDH, family AA3_2) [20], cellobiose dehydrogenase (EC 1.1.99.18) [21], cytochrome P450 [22,23], ring cleavage dioxygenase [24], hydroxylase, monooxygenase [25], hydratase [26], dehydratase [27], demethylase [24], etc. can also participate in the process. Furthermore, various proteins such as electron transfer proteins, receptor proteins, signalling proteins and regulatory proteins are involved in the active transformation of steroid hormones [9]. Thus, an important approach to the determination of the mechanism of oestrogen transformation by a particular microorganism consists of not only the identification of intermediate and final degradation products but also the identification of the key enzymes due to which their transformation occurs.

Fungi of the genus *Trametes* are promising objects for studying oestrogen transformation mechanisms, both from a fundamental and a practical point of view. Their genomes contain a wide variety of laccases and peroxidases as well as other enzymes that may be involved in oestrogen detoxification processes. For example, when annotating the genome of the fungus *Trametes hirsuta* LE-BIN 072, 18 genes encoding peroxidase isoenzymes and 7 genes encoding laccase isoenzymes were discovered [28,29]. Previously, we obtained four laccase isoenzymes (LacA, rLacC, rLacD and rLacF) of *T. hirsuta* LE-BIN 072 [30] in the homogeneous state. When E2 is treated with pure laccase isoenzymes, the oxidative coupling of E2 occurs with the formation of dimers, trimers and tetramers of E2, which are less biologically active than the initial E2 and also form a precipitate, which is easier to remove [31].

The purpose of this investigation was to study the biotransformation of oestrogens by the strain *T. hirsuta* LE-BIN 072 during growth in a medium with E1 and E2, with a detailed study of the fungal secreted proteins (exoproteome) and an assessment of possible transformation products.

## 2. Materials and Methods

### 2.1. Reagents

Commercially available reagents were used: oestrone (3-hydroxyestra-1,3,5(10)-triene-17-one, E1, ≥99%, CAS No. 53-16-7, C_18_H_22_O_2_), 17β-oestradiol (3,17β-dihydroxyestra-1,3,5(10)-triene, E2, ≥98%, CAS No. 50-28-2, C_18_H_24_O_2_) and oestriol (3,16α,17β-trihydroxyestra-1,3,5(10)-triene, E3, ≥97%, CAS No. 50-27-1, C_18_H_24_O_3_) from Sigma-Aldrich (St. Louis, MO, USA) and methanol (HPLC gradient grade, 99.8% CAS No. 67-56-1) from PanReac (Barcelona, Spain). E1, E2 and E3 were used as analytical standards. Other pure, analysis-grade and HPLC-grade materials and solvents were purchased from local commercial suppliers.

### 2.2. Strain and Cultivation Conditions

The basidiomycete strain *T. hirsuta* LE-BIN 072 was obtained from the Collection of Cultures of the V.L. Komarov Botanical Institute (St. Petersburg, Russia).

To obtain the inoculum, the fungus was grown stationary at 25 °C for 10–14 days in 750 mL flasks with glass beads on a glucose–peptone medium (GP) of the following composition (g/L): peptone, 3.0; glucose, 10; KH_2_PO_4_, 0.6; K_2_HPO_4_•3H_2_O, 0.4; MgSO_4_•7H_2_O, 0.5; CaCl_2_, 0.5; MnSO_4_•5H_2_O, 0.05; ZnSO_4_, 0.001; FeSO_4_, 0.0005. The fungus mycelium was destroyed by vortexing with beads for 20 min at 180 rpm until a homogeneous suspension was obtained, which was then sterilely transferred into culture flasks (10% by volume) into a GP medium for liquid-phase cultivation for 7 days at 180 rpm and 25 °C to obtain pellets [32]. Then, wet pellets were washed with sterile water and 10 g was replaced into the flasks with 100 mL of GP (dry biomass weight −1.9 ± 0.1 g). E1 and E2 were added as a solution in methanol (5% vol.) to a final oestrogen concentration of 100 mg/L (370 and 368 μM, respectively).

Samples of culture liquids (CLs) were taken on days 1, 3, 6 and 10. Fungal biomass was separated by filtration and dried at a temperature of 60 ± 5 °C to a constant weight.

### 2.3. Enzymatic Assays

Enzymatic activities were assessed spectrophotometrically on a PerkinElmer Lambda 35 spectrophotometer (PerkinElmer, Shelton, USA) in CLs after the separation of mycelium, as described in [33,34]. Briefly, the total oxidase activities were determined at λ = 436 nm in 0.1 M sodium acetate buffer (pH 4.5), using a solution of 2,2′-azino-bis-(3-ethylbenzthiozolin-6-sulfonic acid) diammonium salt (ABTS) as a chromogenic substrate. The laccase activities were determined at λ = 410 nm in 0.1 M sodium acetate buffer (pH 4.5), using a solution of catechol as a chromogenic substrate. Manganese peroxidase activity was determined by measuring the formation of a Mn^3+^–tartrate complex upon the oxidation of 0.1 mM MnSO_4_ at λ = 238 nm in 0.1 M sodium tartrate buffer (pH 3.0) in the presence of 0.1 mM H_2_O_2_. An increase in optical density in 1 mL of the reaction mixture for 1 min was taken as 1 conventional unit of activity (U). The esterase activity was determined using *p*-nitrophenyl butyrate as a substrate. The reaction was carried out in sodium acetate buffer (pH = 4.5) at 40 °C for 10 min. The reaction was quenched with sodium phosphate buffer (pH = 7.3) and the optical density was determined at λ = 400 nm. Calculation of esterase activity was carried out according to the following formula:A_(U/mL)_ = 0.13 × ΔA_400_ × R_E_(1)
where R_E_ is the preliminary dilution of CL before adding the substrate to the solution.
ΔA_400_ = A_400_ − A_400(S)_ − A_400(E)_(2)
where A_400(S)_ is the control, in which water was used instead of CL, and A_400(E)_ is the control without adding a substrate to the reaction mixture.

### 2.4. Antioxidant Capacity and Phenolic Content

The antioxidant capacity (AOC) in CLs was determined by the oxygen radical absorbance capacity fluorescence method (ORAC) using a Synergy 2 microplate photometer–fluorometer (BioTek, Boston, MA, USA), as described in [35]. This technique relies on the ability of antioxidants to inhibit peroxyl radical (ROO•) oxidation, which can be evaluated by the loss of fluorescence intensity, during ROO• damage. The peroxyl radical was generated directly in the reaction medium during the thermal decomposition of the azo compound 2,2′-azobis (2-methylpropionamidine) dihydrochloride (AAPH, Sigma-Aldrich, St. Louis, MO, USA), initiated by incubation at 37 °C for 10 min. The antioxidant capacity was expressed as the amount of Trolox (Sigma-Aldrich, St. Louis, MO, USA) molar equivalents per mL of CL (μM TE).

The total phenolic content in CLs was determined by the Follin–Ciocalteu method [36]. First, 100 µL of sample was mixed with 500 µL of 10% Follin–Ciocalteu solution and incubated for 5 min. Afterward, 400 µL of Na_2_CO_3_ was added and incubated for 60 min in the dark. The absorbance was measured at 765 nm using a PerkinElmer Lambda 35 spectrophotometer (PerkinElmer, Shelton, USA), while the calibration curve was drawn using gallic acid with a concentration variation of 20–300 ppm. Finally, the total phenolic content was expressed in mg of gallic acid equivalent per mL of CL (μg GAE/mL).

### 2.5. Extraction of Transformation Products

The CLs were separated from the mycelium by filtration through a nylon filter; the mycelium was washed with sterile water to remove residual CL. The rinsing water was combined with the CL. Steroids were extracted from CL and mycelium samples separately:

(1) The extraction from the CLs was carried out with an equal volume of dichloromethane (DCM) three times. The extracts were combined and evaporated to dryness on a Heidolph rotary evaporator (Schwabach, Germany).

(2) The extraction from the mycelium samples was carried out with 10 mL of methanol and then with a mixture of dichloromethane/methanol at a ratio of 1/1 twice. The extracts were combined and evaporated to dryness.

The dry residue after evaporation was dissolved in methanol and then used for analysis.

### 2.6. High-Performance Liquid Chromatography with Mass Spectrometric Analyses (HPLC-MS)

The quantitative assessment of the oestrogen biotransformation degree during the cultivation of *T. hirsuta* and identification of the formed metabolites was carried out by HPLC-MS on an Impact II high-resolution quadrupole-time-of-flight mass spectrometer (Bruker Daltonik, Bremen, Germany) equipped with an Apollo II electrospray ionization source (Bruker Daltonik, Bremen, Germany) and a UHPLC Elute (Bruker Daltonik, Bremen, Germany) on the reverse-phase column Acquity HSS T3 1.8 µm 2.1 × 100 mm (Waters, Manchester, Ireland). Calibration curves were also plotted using the starting substrates E1 and E2 to assess their quantitative content. The quantitative analysis was carried out under the following conditions: flow rate 0.25 mL/min with post-column split 1:20, gradient elution from 5% to 95% B in 10 min (A: 0.1% solution of formic acid in water, B: 0.1% solution of formic acid in acetonitrile), column temperature 30 °C, injection volume 5 µL, electrospray ionization in positive ion mode, capillary potential 4.5 kV, atomizing gas nitrogen 0.9 bar, drying gas nitrogen 5 L/min 180 °C, scanning range *m*/*z* 100–2200, scanning frequency 1 Hz, automatic internal calibration using sodium trifluoroacetate solution. The spectra were processed using the Compass DataAnalysis 5.1 software package (Bruker Daltonik, Bremen, Germany).

Signals for quantification:

E1: *m*/*z* 271.1693 ± 0.01 [M + H]+, RT 8.7 min

E2: *m*/*z* 273.1849 ± 0.01 [M + H]+, RT 8.1 min

Transformation product analysis was carried out under the following conditions: flow rate 0.25 mL/min with post-column flow split 1:20, gradient elution from 2% to 95% B in 15 min (A: 0.1% solution of formic acid in water, B: 0.1% solution of formic acid in acetonitrile), column temperature 30 °C, injection volume 20 µL, electrospray ionization in positive and negative ion modes (separate analyses), capillary potential 4.5 kV (positive ion mode) and 3.5 kV (negative ion mode), atomizing gas nitrogen 0.9 bar, drying gas nitrogen 5 L/min 190 °C, scanning range *m*/*z* 20–1500, scanning frequency for full spectrum 8 Hz, automatic second-order spectra recording mode (collision activation) with a dynamic recording frequency of 6–12 Hz, collision gas nitrogen, collision energy of 20 and 50 eV, automatic internal calibration using sodium trifluoroacetate solution. The processing of the spectra, identification of components, and semi-quantitative analysis were carried out using the Metaboscape 4.0.4 software package (Bruker Daltonik, Bremen, Germany).

Signals for reference compounds:

E1: *m*/*z* 270.1623 ± 0.01 [M]^+^, RT 18.9 min

E2: *m*/*z* 272.1778 ± 0.01 [M]^+^, RT 17.6 min

E3: *m*/*z* 288.1717 ± 0.01 [M]^+^, RT 13.8 min

### 2.7. Two-Dimensional Gel Electrophoresis and MALDI-TOF/TOF MS Identification

The extraction of secreted proteins, sample preparation, 2D gel electrophoresis (*2-DE*) and MALDI-TOF/TOF MS/MS analysis were performed as previously described in [37].

### 2.8. Genome Analysis

The genome sequence of *T. hirsuta* LE-BIN 072 was obtained from the NCBI database (accession number GCA_001302255.2). An annotation of *T. hirsuta* LE-BIN 072 proteins is presented in Appendix A.

### 2.9. Statistical Data Analysis

All measurements were performed in three biological replicates. Whenever appropriate, the measurement was performed in three technical replicates. Results are presented as the mean ± standard deviation. Statistical data processing was carried out by the analysis of variance method (ANOVA). When a significant value (*p* < 0.05) of the F-statistic was found, the differences between individual means were assessed using the Tukeys HSD (honestly significant difference) test (*p* ≤ 0.05). All statistical data manipulations were performed using the R statistical package.

## 3. Results

### 3.1. The Effect of Oestrogens on the T. hirsuta Growth

The ability of the white rot fungus *T. hirsuta* LE-BIN 072 to grow in the presence of oestrogens was assessed by the values of fungal biomass accumulation during the growth in GP + E1 and GP + E2 compared to the control medium (Figure 2).

Oestrogens were added to the medium at a final concentration of 100 mg/L in the form of a solution in methanol (5% vol.). The effect of the presence of a solvent in the medium on the growth and vital activity of the fungus was taken into account during the analysis. Thus, the GP medium and GP medium with methanol (GP + Met, 5% vol.) were used as controls.

Since E1 and E2 are poorly soluble in water due to their lipophilic structure [18,38], the introduction of oestrogens into the aquatic biotransformation medium is usually carried out with solvents. As is known, methanol is one of the products of spruce and birch wood lignin destruction by lignin-degrading fungi [39] and can also be a product of the demethoxylation of veratryl alcohol (3, 4-dimethoxyenezyl alcohol) synthesised de novo from glucose by the fungus during secondary metabolism [40]. Thus, methanol as a solvent for steroid substrates is considered preferable since the extracellular enzyme complex of the white rot fungus is able to utilise it as a carbon source [41].

The results of the experiment showed that the growth of *T. hirsuta*, both in the presence and absence of oestrogens, continued for up to 6 days of cultivation. The addition of methanol and oestrogens to the GP medium slowed down the growth of the fungus. The growth rate of *T. hirsuta* fungal biomass in GP + Met, GP + E1 and GP + E2 decreased and amounted to 0.52, 0.11 and 0.41 g/day, respectively, compared with the GP medium, which was 0.76 g/day. The inhibitory effect of methanol combined with oestrogens was more pronounced: if by day 6 the amount of biomass in the presence of methanol alone increased by approximately 2.5–3 times, then in the presence of E1 and E2, it increased by 1.4 and 2.2 times, respectively, compared with the initial biomass. The inhibitory effect of steroid hormones on fungal growth *in vitro* is known [41].

Thus, it was shown that *T. hirsuta* LE-BIN 072 is able to grow on a medium with methanol and oestrogens. In addition, the results obtained are generally consistent with known data published in studies [42]. This, it is possible to use methanol to introduce steroid substrates in concentrations of up to 5% during microbial transformations.

### 3.2. Enzyme Activities

The white rot fungi, in particular the primary wood-destroying saprotroph *T. hirsuta*, having in their arsenal a wide range of nonspecific extracellular enzymes (including laccases, peroxidases and esterases), are capable of decomposing a broad range of EDCs [37,43,44]. In this regard, the changes in these enzymatic activities during the growth of the fungus in the presence of oestrogens were studied (Figure 3).

The maximum total oxidase activity with ABTS as a substrate (TOA/ABTS) of the strain grown on the control GP medium reached 350 U/L by day 6 (Figure 3a). However, surprisingly, in the presence of methanol and methanol with oestrogens, the TOA/ABTS was completely absent. At the same time, the total oxidase activity with catechol as a substrate (TOA/catechol) of *T. hirsuta* gradually increased up to day 10 (Figure 3b) during growth on the control GP medium. However, in the other media, the maximum TOA/catechol was observed as early as day 3, and on the GP + E2 medium, the activity significantly exceeded the activity of the other samples (up to 10 U/L). In contrast, the activity on the GP + E1 medium was minimal. Thus, in the presence of E2, the maximum TOA/catechol was 11 times higher than in the presence of E1. The presence of the peroxidase activity was also shown in the studied CLs using Mn^2+^ as substrate (Figure 3c). The pattern of changes in the activities during the cultivation was similar regardless of the medium and had two peaks—on the 3rd day and on the 10th day of cultivation. In this case, the maximum activity was also achieved on the 3rd day. The maximum manganese peroxidase activity was 1.7 times higher in the presence of E1 compared to E2 on the 3rd day of cultivation. Similar oxidative enzyme activity profiles in the fungal CLs with the peaks and subsequent decreases to trace values were previously shown during the cultivation of *T. hirsuta* LE-BIN 072 on media such as GP in the presence of CuSO_4_ and GP supplemented with lignocellulose substrate—milled oat straw [45]. As for esterase activity, a noticeable increase in this activity was also observed on the 3rd day of cultivation in the presence of oestrogens, in contrast to GP and GP + Met (Figure 3d). The maximum activity values were noted on the 6th day of cultivation, and, for the GP + Met and GP + E1 samples, the activity was significantly higher than for GP. On the 10th day, only the activity on GP and GP + E1 did not change significantly, while, for the other environments, on the contrary, it decreased.

### 3.3. Antioxidant Capacity and Total Phenolic Content

The inability to detect the TOA/ABTS in the fungal CLs with methanol was presumably associated with an increased amount of antioxidants formed in response to the appearance of methanol in the medium. Therefore, to test this hypothesis, the antioxidant capacity (AOC) and total phenolic content were measured in CLs (Figure 4).

It was shown that in the presence of methanol alone or with oestrogens in the medium, the antioxidant activity and the total phenol content in CLs significantly increased. Their maximum values were reached on the 3rd and 6th day of cultivation. In the control GP medium, the antioxidant activity decreased with time, and the content of phenols did not change significantly.

The high values of antioxidant activity we discovered in GP + Met, GP + E1 and GP + E2 correlated with the absence of oxidase activity in these samples, when ABTS was used as a substrate. This can be explained by the reduction of ABTS•+ radical cations by the antioxidants present in the medium [46], such as, for example, phenolic compounds, the increased content of which was also found in GP + Met, GP + E1 and GP + E2. Phenolic compounds are one of the most important natural antioxidants. It is known that phenolic acids as well as flavonoids are the most common phenolic compounds found in fungi and responsible for their biological activity, including their antioxidant activity [47].

Thus, the positive correlation between AOC and the total phenolic content in CLs indicates that phenolic compounds may contribute to the antioxidant capacity of these fungi. It is important to note that the formation of free radicals that can cause oxidative stress plays an important role in the mechanisms of xenobiotic detoxification by fungi, including different aromatic compounds. At the same time, the maximum antioxidant activity, detected on the 3rd day of cultivation (Figure 4), correlated with the maximum (at the same time point) TOA/catechol and peroxidase activities, the action of which ensured the formation of free radicals.

### 3.4. Biotransformation of Oestrogens by T. hirsuta LE-BIN 072

The efficiency of the oestrogen biotransformation by the white rot fungus *T. hirsuta* was defined as the residual amount of the E1 and E2 in the extracts of culture liquid and mycelium relative to the amount of initial oestrogens in the growth medium. The experiment showed that after 24 h of cultivation, the total residual amount of both E1 and E2 was about 8% and did not change significantly during further cultivation for E1. For E2, trace amounts were detected, starting from day 3 (Table 1, columns Total).

Similar results were obtained by Ge et al. in [48], where it is shown that the degradation efficiency of E2 (10 μM) by *Candida utilis* CU-2 was 92% and 97% during 2 and 6 days, respectively. The complete removal of E2 (0.1 μM) by *Phanerochaete chrysosporium* after 5 days was demonstrated in [11]. Tamagawa et al. also tested the degradation of E1 (10^−4^ M) by the ligninolytic fungus *Phanerochaete sordida* and recorded 95% transformation within 5 days of incubation [13]. The possibility of removing 97% of E2 (10 mg/L initially) in 24 h by the fungus *Trametes versicolor* was also shown [49]. However, it should be noted that the initial concentration of oestrogens in the described studies was significantly lower than in this work (initial E1 and E2 content was 100 mg/L or 370 and 368 µM, respectively). Thus, the ability of the fungus *T. hirsuta* to grow and effectively transform oestrogens in high concentration indicates the resistance of this strain to the toxic effect of oestrogens, which is promising in terms of using this strain for mycoremediation.

It is known that oestrogens can be partially adsorbed on mycelium during transformation. For example, in [49], it was described that when E2 (10 mg/L) was transformed with *T. versicolor* mycelium, 97% of the initial substrate was removed after 7 days. At the same time, when E2 was incubated with autoclaved mycelium (inactive) under the same conditions, 38% of E2 was removed due to sorption. Therefore, in our study, the content of E1 and E2 during the cultivation was assessed separately in CL extracts and mycelium extracts by using the HPLC-MS method (Table 1).

From the data presented in Table 1, it follows that more than 65% of the residual E1 and E2 were found in mycelium extracts. This allowed us to conclude that during the transformation of E1 and E2 by *T. hirsuta,* there is a high sorption degree of substrates on the mycelium due to the hydrophobicity of oestrogen molecules.

It is important to emphasize that when identifying metabolites by the HPLC-MS analysis, the presence of the product E2 in the GP + E1 medium and the product E1 in GP + E2 was shown. Thus, we can conclude that E1 can be formed from E2, and vice versa. Quantitative analysis showed that the E2 content in the GP + E1 medium significantly exceeds the E1 content in the GP + E2 medium. Figure 5 shows the accumulation curves of E1 and E2 as transformation products.

In addition, a compound with MW 288 assigned to oestriol (E3, C_18_H_24_O_3_) was detected in CL extracts on the 1st day. The identity of this compound was confirmed by comparison with the commercial reference standard E3. Other potential transformation products are proposed based on our HPLC-MS results and literature data (the data are presented in Table 2).

It is important to note that most of the metabolites found were present in both CLs and mycelial extracts (Table 2). According to the results, the dominant product of the E1 and E2 transformation was the compound with a calculated *m*/*z* [M]^+^
*=* 578.2857 (Rt 25.67). This compound corresponded to the dimer without designation of the structure (C_36_H_34_O_7_) that we had previously obtained during the transformation of E2 with pure *T. hirsuta* laccases [31]. In addition, there were signals with calculated *m*/*z* [M]^+^
*=* 538.3064, 540.324 and 542.341 corresponding to the E1-E1 dimers (C_36_H_42_O_4_), E2-E1 dimers (C_36_H_44_O_4_) and E2-E2 (C_36_H_46_O_4_). These compounds were detected throughout the entire cultivation period on GP + E1 and GP + E2 both in the CL extracts and in the mycelium extracts. The exception was the E2-E2 dimer, which appeared later on the GP + E1 medium (on the 3rd day) in the CL extract and then on the 6th day it was identified in the mycelium. Dimeric quinone-like derivatives were also found. The detected signals corresponded to (2-OH-E1)-E1 and (4-OH-E1)-E1 (calculated *m*/*z* [M]^+^ = 554.263, C_36_H_42_O_5_) and (2-OH-E2)-E2 and (4-OH-E2)-E2 (calculated *m*/*z* [M]^+^ = 558.3358, C_36_H_46_O_5_). Obviously, the formation of dimers occurred outside the cell, after which, they were sorbed on the mycelium, which subsequently allowed them to be identified in the mycelium extracts.

It should be noted that minor metabolites **3**, **7, 14** (or **15**) and **18** were present only in CLs and metabolites **23** (or **29**) and **31** were present only in mycelial extracts (Table 2).

### 3.5. Compositional Analysis of T. hirsuta LE-BIN 072 Exoproteomes in the Presence of Oestrogens

For the exoproteomic study, samples of CLs at 1, 2, 3, 6 and 10 days of cultivation were collected. Combined samples from days 1, 2 and 3 (1 + 2 + 3 days) and separate samples from the 6th and 10th days were used for analysis. Proteins were extracted and analysed by 2D electrophoresis (protein maps are presented in Appendix A). The individual presence of identified proteins is depicted as an UpSet-style plot in Figure 6. The collective presence of the proteins is depicted in the Venn diagrams (Figure 7).

A total of 29 different proteins were detected in the samples (Figure 6 and Appendix A), 28 of which were identified by MALDI-TOF/TOF MS/MS analysis. The spectrum and ratio of proteins in the exoproteomes changed depending on the duration of cultivation and the culture medium (Figure 7). Moreover, more significant differences in the protein spectrum were found for different days of fungal cultivation than for different cultivation media. Thus, in the combined samples (1 + 2 + 3 days) in all three media, the main share of the total pool of proteins was represented by ligninolytic peroxidases (two manganese peroxidase (EC 1.11.1.13)—MnP5 (p1, FUN_002850) and MnP7 (p2, FUN_007983)—and one versatile peroxidase (EC 1.11.1.16)—VP2 (p3, FUN_007604)) (Figure 6 and Appendix A). This result correlated well with the results for enzymatic activity in CLs on the 3rd day (Figure 3b,c). Mn^2+^ ions and catechol are substrates for both MnPs and VPs, but the latter oxidize phenolic substrates like catechol much better. Indeed, on the medium with oestrone (GP + E1), the oxidative activity measured for the substrate Mn^2+^ on the 3rd day was higher than for GP + E2. On the contrary, on the medium with oestradiol (GP + E2), the oxidase activity on the 3rd day was higher for the substrate catechol. This was consistent with the data for the exoproteome analysis: in GP + E1, the share of secreted MnPs prevailed among the ligninolytic peroxidases compared to VP2; in GP + E2, on the contrary, the share of VP2 was significantly higher.

In addition to ligninolytic peroxidases, two isoenzymes of glucose–methanol–choline (GMC) oxidoreductase were detected in all media: FUN_010541 (p4) and FUN_004737 (p5). Moreover, p4 was stably detected throughout the entire cultivation period, while p5 was detected only on the 10th day of cultivation. The amount of p4 in the secretomes changed depending on the medium used (Appendix A). Thus, on the 3rd and 10th days of cultivation, the amount of p4 on the GP + E2 medium was lower compared to other media. On the 6th day, on the contrary, there was more p4 in the presence of E2. Comparative analysis of FUN_010541 (p4) and FUN_004737 (p5) amino acid sequences with available published amino acid sequences of putative fungal GMC oxidoreductases using the BLAST algorithm showed 80.51% and 89.41% similarity (%Identity) with pyranose dehydrogenase from *Trametes pubescens* (OJT14639.1) and alcohol oxidase from *Trametes meyenii* (KAI0644068.1), respectively, belonging to the Aryl-alcohol oxidase (AAO–PDH) cluster [57]. Thus, p4 and p5 were assigned to the AAO–PDH cluster. It should be noted that AAOs were not detected in significant quantities in the *T. hirsuta* secretomes obtained earlier on the GP medium [45,58]. It is obvious that the presence of methanol in the culture medium, either alone or with oestrogens, induced the production of AAOs.

Three isoenzymes of the carboxylesterase PnbA family were identified in the secretomes: p8 (FUN_010034), p9 (FUN_007692) and p10 (FUN_005650). Protein p8 appeared earliest in the sample 1 + 2 + 3 on GP + E1 and was then present on all days in all media. Carboxylesterase p9 appeared on the 6th day in all media. In addition, carboxylesterase p10 was detected in the sample 1 + 2 + 3 only in GP + E2 and on the 10th day only in GP + E1. The activity of these enzymes may be responsible for the observed increase in total esterase activity in fungal CLs from day 3, with the maximum activity on the 6th day, when both carboxylesterase isoenzymes, p8 and p9, were detected in all samples (Figure 3d and Figure 7). The noticeable induction of carboxylesterase p8 on the 6th day in the GP + E2 medium (Appendix A), compared with other media, corresponded to the maximum esterase activity shown for this medium at this time point (Figure 3d). Carboxylesterases are important components of xenobiotic-metabolizing enzyme (XME) complexes in a variety of organisms. These enzymes perform a variety of functions, including catalysis of hydrolytic reactions of carboxyl esters, phosphate esters and other chemical compounds, and exhibit diverse substrate selectivity due to their spacious and adaptive binding pocket. These enzymes efficiently bind equivalent amounts of noxious chemicals, resulting in xenobiotic sequestration [59].

Lytic polysaccharide monooxygenase (LPMO) FUN_000419 (p7) was found only in the combined sample 1 + 2 + 3 in all media, and glyoxal oxidase (GLOX) FUN_010480 (p6) was primarily secreted in GP + E1 on days 1 + 2 + 3 and was detected in all samples only by day 10. Alpha-amylase FUN_010348 (p12), glucanosyltransferase FUN_000340 (p22) and necrosis-inducing secreted protein FUN_007129 (p27) were also detected in samples 1 + 2 + 3 and were stably detected throughout the entire cultivation period. In addition, the combined sample 1 + 2 + 3 contained an additional unique isoenzyme alpha-amylase FUN_010349 (p13).

Interestingly, the spectrum of secreted proteins narrowed by the 6th day of cultivation (12 proteins in total). Ligninolytic peroxidases were not detected in the exoproteomes, while the oxidative activity in the CLs also sharply decreased (Figure 3b,c). The main share of the total protein pool in all media at this time point consisted of alcohol oxidase FUN_010541 (p4), alpha-amylase FUN_010348 (p12), starch binding domain FUN_009362 (p14) and the carboxylesterase PnbA family FUN_010034 (p8). Acid protease FUN_006611 (p26) was present only in the GP + E1 medium. The protein of glycoside hydrolases family 32 (GH32) FUN_007710 (p20) was detected in the GP + Met and GP + E1 only at this time point. In addition, the hypothetical protein FUN_009219 (p29) was present in all samples only on the 6th day.

The spectrum of secreted proteins was widest by the 10th day of cultivation (22 proteins in total). Most of them (17) were detected in all media (Figure 7B). Among the ligninolytic enzymes, only the VP2 protein was detected in all media. Probably, the manganese peroxidase activity detected on the 10th day was due mainly to the secretion of VP2. At the same time, on the 3rd day, the total peroxidase activity was apparently detected for all three peroxidases (MnP5, MnP7 and VP2) using Mn2+ as a substrate. In all exoproteomes on the 10th day, proteins such as alpha-trehalose glucohydrolase (EC 3.2.1.28; α,α-trehalase) FUN_006612 (p21), Cl-channel transport protein FUN_009584 (p26) and laminarinase FUN_002148 (p17) appeared. In the GP + E1 medium, four unique proteins were additionally detected: carboxylesterase FUN_005650 (p10); laminarinase FUN_001432 (p16); chitinase FUN_007364 (p18) and melibiase FUN_000432 (p19), which was also present in the GP + E1 medium in the combined sample (1 + 2 + 3) and disappeared on the 6th day. The appearance in GP + E1 and GP + E2 of proteins involved in the remodelling and transformation of fungal cell walls (glycoside hydrolase GH16), as well as the appearance on GP + E1 of chitinase GH18 and intracellular α,α-trehalase involved in the hydrolysis of trehalose (the main storage disaccharide of fungi), may have indicated the onset of cell lysis. This was consistent with the biomass growth curves (Figure 2). In media with oestrogens, a decrease in mycelium biomass was observed by day 10, after the stationary growth phase.

Thus, in the first 3 days of cultivation, a significant induction of the ligninolytic peroxidase and aryl-alcohol oxidase secretion by the fungus *T. hirsuta* was observed with a simultaneous decrease in the amount of oestrogens in the CLs. Subsequently, the number of carboxylesterases (on the 6th day) and hydrolases (on the 10th day) of the fungal cell wall increased in exoproteomes.

## 4. Discussion

The results of the present study show that by the 3rd day of culturing *T. hirsuta* LE- BIN 072 in the presence of oestrogens, the total residual amount of the initial E1 and E2 was less than 10% (Table 1). At this time point, the activity of oxidative enzymes was at the maximum (Figure 3), and three ligninolytic peroxidases were identified in the secretomes—MnP5, MnP7 and VP2. On the 6th day, peroxidases were absent in the exoproteomes (Figure 6), which correlated with the absence of oxidative activity in the CLs (Figure 3b,c). Moreover, these proteins constituted a larger proportion of the total protein pool in the exoproteome on the 3rd day (Appendix A). We previously demonstrated that these peroxidases differ in their substrate specificity and are involved in the degradation of lignin by the fungus *T. hirsuta* LE-BIN 072 [60]. Thus, peroxidases MnP5 and VP2 are less specific compared to MnP7, and their secretion begins earlier in the course of lignin destruction. At the same time, the activity of VP2 on phenolic substrates is higher than that of MnPs [61]. Interestingly, in the present study, the proportion of MnPs in the secretome was higher in the presence of E1, while, in the presence of E2, on the contrary, VP2 prevailed (Appendix A). This may indicate that E1 (or its degradation products) are likely to induce MnP expression to a greater extent, whereas E2 (or its degradation products) are likely to induce VP2 expression. In [61,62], the ability to remove E2 with a pure commercial preparation of lignin peroxidase and versatile peroxidase from the ligninolytic fungus *Bjerkandera adusta* was demonstrated; however, the metabolites were not identified. In this regard, it is obvious that establishing the specificity of MnP5, MnP7 and VP2 to certain compounds will require obtaining these enzymes in pure form in the future.

In addition to ligninolytic peroxidases, exoproteomes contained elevated levels of aryl-alcohol oxidases (AAOs), whose substrates are a wide range of aromatic and aliphatic alcohols, which are oxidized to the corresponding aldehydes and ketones with the oxidative generation of hydrogen peroxide [63]. The hydrogen peroxide formed during these reactions is, on the one hand, a co-substrate for reactions involving peroxidases, while, on the other hand, H_2_O_2_ participates in Fenton reactions with the formation of hydroxyl radicals, which can also participate in the extracellular transformation and destruction of E1 and E2. The increased secretion of AAOs in the presence of oestrogens correlates with an increase in the content of phenols in these media and an increase in antioxidant activity. Thus, the presence of methanol in the medium alone or in combination with E1 and E2 causes oxidative stress and leads, in turn, to a decrease in the growth of fungal mycelium, and, in the case of a GP + E1 medium, the effect of the inhibition of biomass growth is stronger compared to GP + E2.

The absence of laccases among the identified secreted proteins was an unexpected result for us since a number of studies have shown that laccase is induced by the presence of oestrogens in media [13,64,65]. The formation of dimeric, trimeric and tetrameric products of the oxidative coupling of E2 catalyzed by pure laccases of *T. hirsuta* LE-BIN 072 was shown by us earlier in the article [31]. The fungus *T. hirsuta* LE-BIN 072 previously produced the major isoenzyme LacA in varying quantities on a GP medium. Thus, the absence of LacA in the control GP + Met medium as well as GP + E1 and GP + E2 was obviously caused by the presence of methanol. Despite the fact that the dominant products of oestrogen transformation in the present experiment were dimers (Appendix A), compounds that could be classified as trimers and tetramers were not found among the products. The absence of trimers and tetramers can probably be explained by the high content of alcohol oxidases throughout the cultivation. It is known that *Pleurotus ostreatus* alcohol oxidase prevents lignin repolymerisation during its degradation [66]. Based on the totality of the data obtained, it was concluded that extracellular peroxidases apparently play a key role in the dimerization process, quickly oxidizing the phenolic functional groups of oestrogens to form phenoxyl radicals, which then combine with each other to form the corresponding dimers with lower toxicity compared to the initial substrates [11]. For example, Figure 8 shows the structures of E1-E1, E2-E2 and E2-E1 possible dimers.

Also, minor compounds of E1 and E2 transformation with MW 288, assigned to 4-OH-E2 (**12**) or 2-OH-E2 (**13**) (steroidal catechols), their possible 17-keto analogues 4-OH-E1 (**16**) or 2-OH-E1 (**17**) with MW 286, as well as compounds with MW 304 and MW 302, assigned to 2,4-diOH-E2 (**18**) and 2,4-diOH-E1 (**19**), respectively (steroidal pyrogallols), were identified [48,50,51]. This suggested that the detoxification of oestrogens by white rot fungi may occur with the formation of catechol- and pyrogallol-like derivatives E1 and E2 by the hydroxylation of the phenolic ring of the steroid molecule (Figure 9 and Figure 10).

Phenolic oestrogen derivatives (including E1 and E2), especially catechol-like ones, are known to function as antioxidants and free radical scavengers under various experimental conditions [67]. Thus, the formation of these products may also have contributed to the increase in antioxidant activity that we observed when culturing the fungus on media containing oestrogens. Note that the processes of oxidation–reduction of the 17-oxygen function at C17 of the steroid molecule can occur both before and after the formation of catechol-like compounds [51], which can subsequently undergo dimerisation. In this case, intermolecular interaction proceeds by the mechanism of radical single-electron transfer (SET) [68]. Radicals can interact with each other to form various isomeric dimers (for example, (2-OH-E1)-E1 and (4-OH-E1)-E1 (MW 554, C_36_H_42_O_5_) and (2-OH-E2)-E2 and (4-OH-E2)-E2 (MW 558, C_36_H_46_O_5_)).

Among the minor metabolites found, the compound E3 (**3**, MW 288.1717, C_18_H_24_O_3_) was identified only in the CL extracts on the 1st day. Zhou et al. previously showed the formation of this metabolite during the degradation of E2 by the fungus *P. chrysosporium* [11], but not in all the media used. So, in potato medium, E1 and E3 (less active metabolites than the initial E2) were formed as the main intermediates. In this case, E2 was first transformed into E1 and then into E3 before further degradation. However, in Kirk’s medium, only a small amount of E1 was formed as, in these conditions, the release of extracellular peroxidases (LiP and MnP) occurred. In this case, oestrogens were rapidly oxidised by peroxidase to form phenoxyl radicals, which then underwent oxidative coupling. It should be noted that in our experiment, on day 3 and later, E3 was no longer detected in the CLs, while the products of oxidative coupling E1 and E2 were detected throughout the entire cultivation period (Table 2). Figure 3b,c demonstrate that on the 1st day, the peroxidase activity was only beginning to increase, whereas, by day 3, it had already reached its maximum. Probably for this reason we were able to find the metabolite E3 only in the extracts obtained on the 1st day. In the work [69], the pathway of E1 (**1**) conversion into E3 (**3**) via a 16α-hydroxyestrone (16-OH-E1, MW 286, C_18_H_22_O_3_) intermediate (Figure 9) is described. We also detected signals with *m*/*z* [M + H]+ = 287 (in CL and mycelial extracts on the 1st day of cultivation and only in mycelial extracts on the 3rd day), which were assigned to 16α-hydroxyestrone (**4**). This compound could be formed as a result of the hydroxylation of E1 (**1**), which was carried out by steroid hydroxylases (cytochrome P-450-dependent monooxygenases) that were localized in the endoplasmic reticulum of the cells [70]. Using EggNOG-Mapper’s algorithm, 143 genes encoding cytochrome P450 family proteins were predicted in the *T. hirsuta* LE-BIN 072 genome [28], with 6 of them containing a signal peptide (FUN_001652, FUN_002069, FUN_004831, FUN_005937, FUN_008108, FUN_008195), i.e., being externally secreted proteins (Appendix A). However, in this study, these proteins were not detected in the exoproteome. Perhaps they can be associated with the cell wall. Product **4** could also be formed from compound E3 by oxidation of the hydroxyl group at C17 due to intracellular dehydrogenation processes by the enzyme 17β-HSD [50], which are also annotated in the *T. hirsuta* genome (FUN_007585 and FUN_008591) (Appendix A). These same enzymes can be involved in the conversion of E1 to E2 and vice versa, confirmed in our work.

We also suggest the possibility of formation of the 4-methoxy and 2-methoxy derivatives of E2 and E1 (**14** or **15** and **20** or **21**, respectively) (Figure 9 and Figure 10). It is known that oestrogens containing hydroxyl groups in the A ring can be converted into methoxy derivatives by the action of catechol O-methyltransferases (COMT, EC 2.1.1.6) [56,69]. It should be noted that COMT were found in the mycelia of the wood fungus *Lentinula edodes* [71]. Analysis of the *T. hirsuta* genome revealed 97 methyltransferases. Seven of these were putative O-methyltransferases (Appendix A). A deeper analysis of these protein sequences using the BLAST algorithm revealed the O-methyltransferase family 3 protein FUN_005291 (pfam01596), which is a putative catechol-O-methyltransferase.

As mentioned earlier, the biotransformation of steroid compounds by microorganisms can occur through the degradation of the steroid core using the breakdown products as a source of carbon and energy. In this case, the destruction of the cyclopentanoperhydrophenanthrene skeleton begins, as a rule, with the splitting of rings A and D. However, the initiation of the steroid nucleus destruction by cleavage of the B ring cannot be excluded. The precursors of the cleavage of the A ring of oestrogens can be catechol-like derivatives of E2 and E1, namely, compounds **12** and **13** and **16** and **17**, as well as their quinone-like derivatives **22** and **28** (MW 286) and **23** and **29** (MW 284), respectively (Figure 9 and Figure 10). Cleavage of the 1–2 or 3–4 bonds of the aromatic ring A in compound **17**, presumably by extradiol ring-cleavage dioxygenase (ERCD, FUN_005230), can lead to the formation of products **32** or **33**, and, as a result of cleavage of the ring A at the 2–3 and 4–5 bonds in the catechol-like oestrogen **16**, compounds **35** and **36** are formed (Figure 9). Metabolite **36** was previously identified by Sh. Li et al. [72] as a key product of the phenolic ring A cleavage of compound **16** at the 4,5-bond when studying E2 biodegradation metabolic mechanisms by the *Novosphingobium* sp. ES2-1. Quinone-like structures and their metabolites are precursors to the subsequent cleavage of the rings A and B [54]. Possible degradation products of the aromatic ring A may also include metabolites **34** and **37** (Figure 9) [51]. The formation of meta-cleavage metabolite **37** from 4OH-E1 (**16**) was previously observed during the aerobic transformation of E1 by the bacteria *Sphingomonas* sp. strain KC8 [55]. The metabolite with MW 266 (**26**, Figure 10) we detected was previously identified by Wang, Y. et al. as a product of the destruction of the A ring of the E2 molecule [51]. The detected compound **27** with MW 264 probably corresponded to the 17-keto analogue of compound **26**, which could be formed as a result of the action of the 17β-HSD enzyme (Figure 10).

The 15(16)- and 16(17)-dehydrogenated metabolites E1 and E2 (**5, 7** and **10**) can be considered probable key intermediates for the subsequent enzymatic cleavage of ring D (Figure 9 and Figure 10). It could be assumed that the appearance of 16(17)-dehydro-derivatives was due to the dehydration of the 17β-hydroxyl group of E2 under the action of dehydratase. Among the products of E1 and E2 biotransformation, a compound with MW 254, corresponding to the ∆^16(17)^-derivative of E2—oestratetraenol (E0, **5**)—was found as a possible dehydration product of the 17β-hydroxyl group in E2. It was identified as a possible intermediate product of the cleavage of the D ring, formed during the process of E2 dehydration by *Nitrosomonas europaea* [52].

Signals with MW 290 and 292 can presumably be attributed to the products of the destruction of the D ring of E1—compounds **38** and **42**, respectively (Figure 9). A metabolic pathway for the degradation of E2 to form compound **40** via the sequential formation of compounds E1, **38** and **39** was proposed by H.B. Lee and D.Liu [73]. The degradation pathway of the E1 molecule through the formation of intermediate **41** followed by the cleavage of ring D (compound **42**) and further destruction of ring C was proposed by Y. Wang et al. based on metabolite structures that were detected [51].

The detected signal MW 268 can be attributed to the compound 6,7-dehydro-E1 (**11**, Figure 10). The formation of the 6,7 double bond can be considered a prerequisite for the cleavage of the B ring. The formation of the dehydration product of the 6-hydroxylated derivative of E1 during biotransformation in agricultural soil is described by Li Ma and Scott R. Yates [53].

## 5. Conclusions

In summary, the process of E1 and E2 transformation by the fungus *T. hirsuta* LE-BIM 072 proceeds mainly in the direction of the initial substrates’ detoxification—the radical oxidative coupling of E1 and E2, as well as their metabolites in various combinations, with the formation of less toxic dimers. The main contribution to this process is obviously made by ligninolytic peroxidases MnP5, MnP7 and VP2, detected in the exoproteome in dominant quantities on days 1–3 of cultivation, when the content of the initial E1 and E2 in the medium sharply decreased (less than 10%). That is, this fungus is able to effectively detoxify oestrogens in high concentrations, using the same extracellular enzymes that are active in the destruction of lignin, due to their broad substrate specificity. In addition, the destruction of the steroid nucleus of oestrogens with the cleavage of rings A/B and D with the preceding formation of catechol-, pyrogallol- and quinone-like derivatives takes place.

Thus, the primary wood-destroying saprotroph *T. hirsuta* is a promising object for studying the mechanisms of degradation of phenol-like steroids. To establish the detailed mechanism of degradation of oestrogens by white rot fungi by secreted enzymes, it is of interest to obtain pure peroxidases and establish their specificity with respect to oestrogens and their degradation products in the future.

## Figures and Tables

**Figure 1 jof-10-00795-f001:**
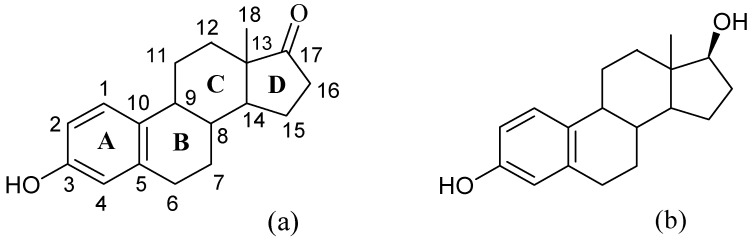
Structures of oestrone E1 (**a**) and 17β-oestradiol E2 (**b**).

**Figure 2 jof-10-00795-f002:**
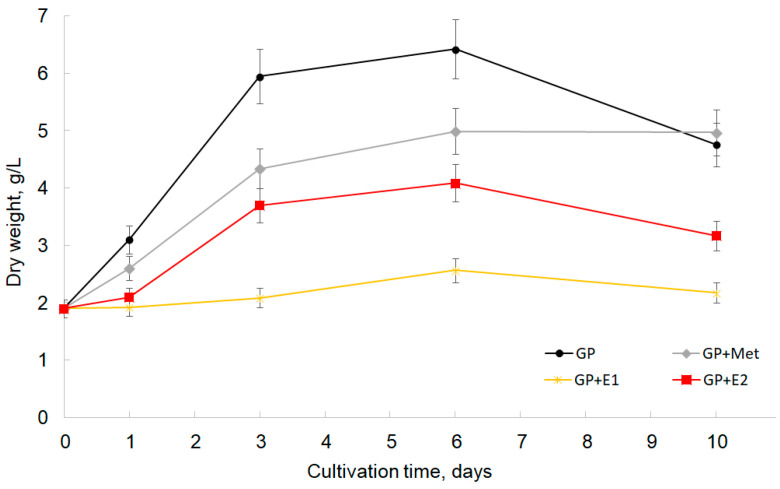
Growth curves of *T. hirsuta* LE-BIN 072 in the GP medium (black line), GP + 5% methanol (grey line, GP + Met), GP + E1 in methanol (yellow line) and GP + E2 in methanol (red line).

**Figure 3 jof-10-00795-f003:**
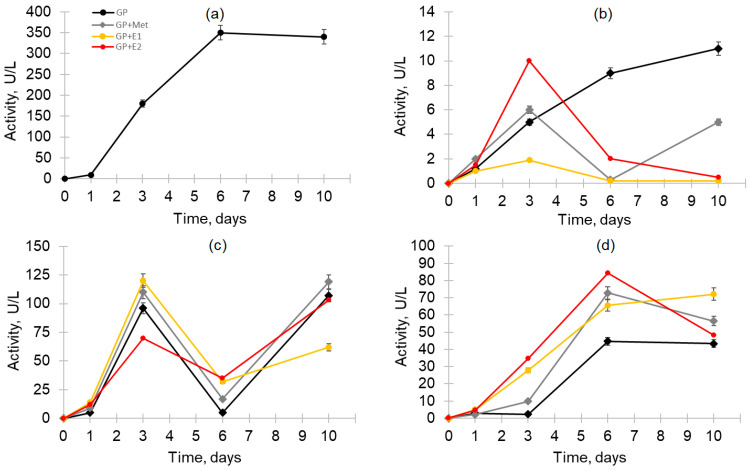
Changes in enzyme activities during the cultivation of *T. hirsuta* LE-BIN 072 in the GP medium (black line), GP + Met (grey line), GP + E1 (yellow line) and GP + E2 (red line): (**a**) total oxidase activity (ABTS used as substrate); (**b**) total oxidase activity (catechol used as substrate); (**c**) manganese peroxidase activity (Mn^2+^—used as substrate); (**d**) total esterase activity (*p*-nitrophenyl butyrate used as substrate).

**Figure 4 jof-10-00795-f004:**
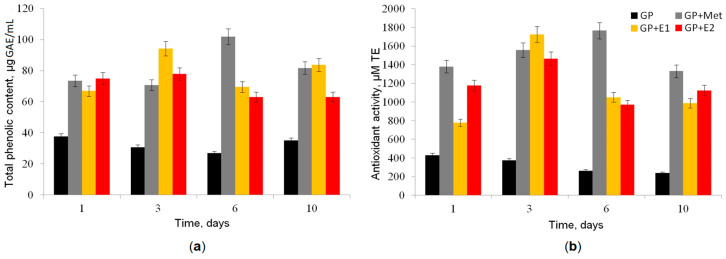
Changes in the total phenol content (**a**) and antioxidant capacity (**b**) of the CLs during *T. hirsuta* LE-BIN 072 cultivation in medium: GP (black bars); GP + Met (grey bars); GP + E1 (yellow bars) and GP + E2 (red bars).

**Figure 5 jof-10-00795-f005:**
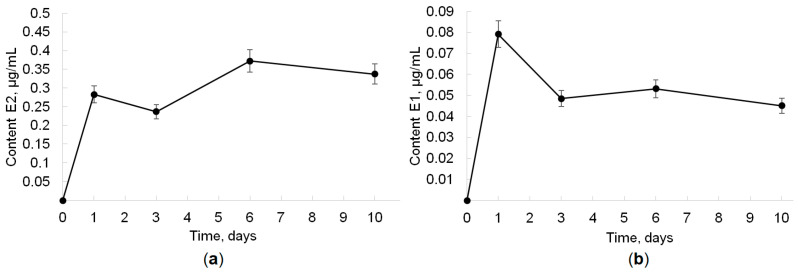
E2 and E1 accumulation as biotransformation products (in culture supernatant and mycelium extracts): (**a**) E1 is initial substrate (GP + E1 medium); (**b**) E2 is initial substrate (GP + E2 medium).

**Figure 6 jof-10-00795-f006:**
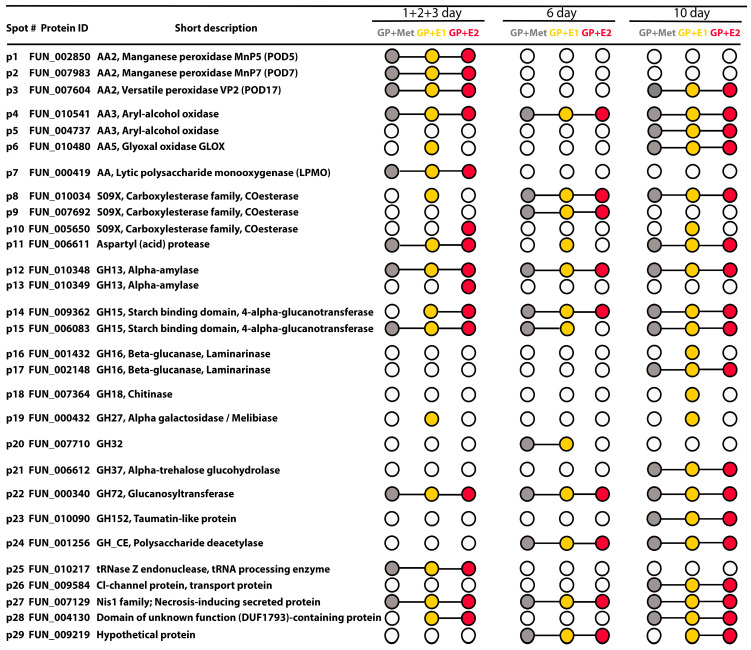
List of proteins secreted by *T. hirsuta* LE-BIN 072 during oestrogen transformation. The presence of proteins in CLs on the corresponding day is indicated by a coloured circle; if the protein was detected in several media, the corresponding circles are connected by a solid line. Presence is indicated in grey for GP + Met, yellow for GP + E1 and red for GP + E2.

**Figure 7 jof-10-00795-f007:**
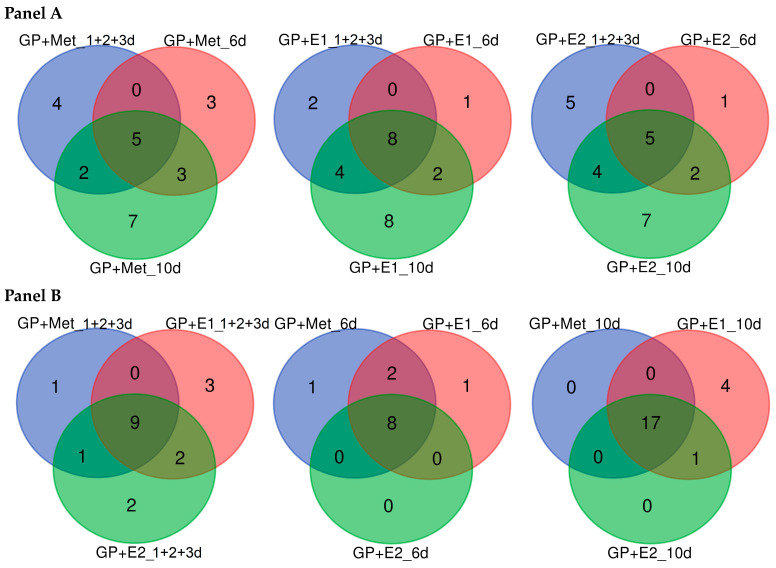
Venn diagrams of collective presence of secreted proteins: panel (**A**) shows the change in proteins for each sample during cultivation; panel (**B**) shows the differences between samples at 1 + 2 + 3, 6 and 10 days.

**Figure 8 jof-10-00795-f008:**
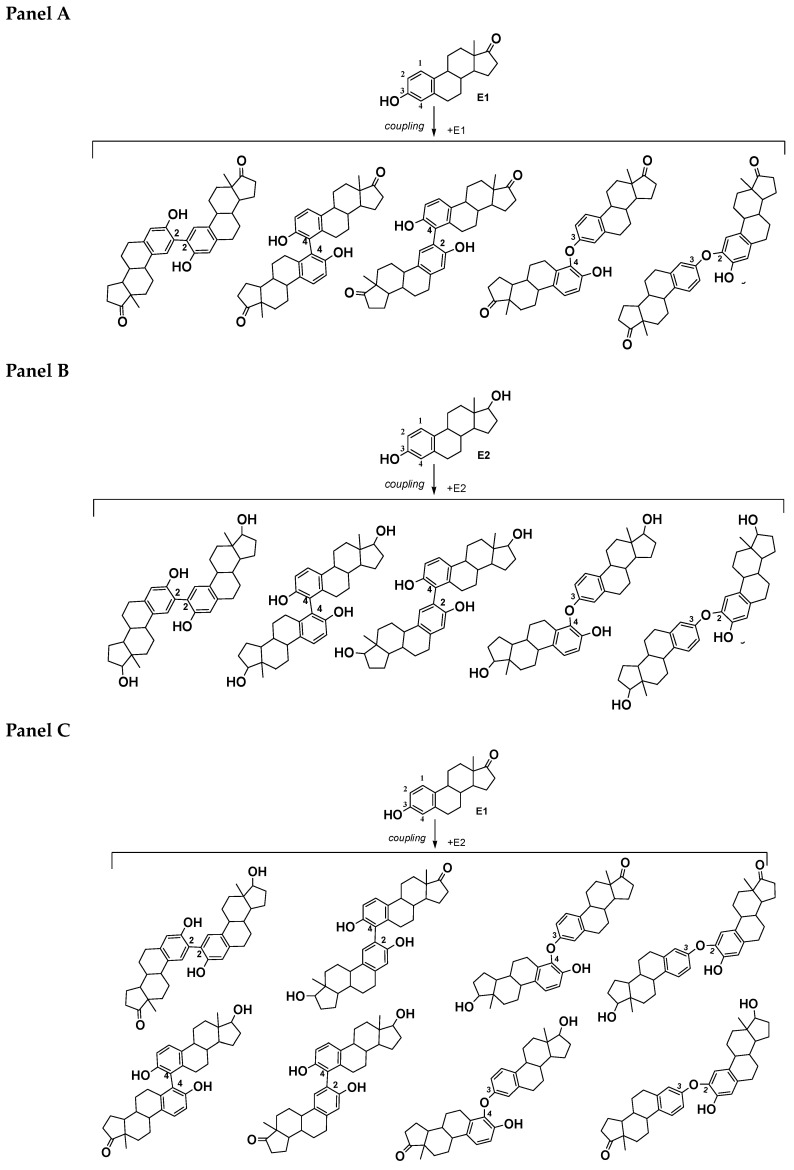
Proposed scheme of formation of possible dimeric products of the E1 and E2 oxidative coupling catalysed by *T. hirsuta* LE-BIN 072 peroxidases (PODs): panel (**A**) E1-E1 dimers; panel (**B**) E2-E2 dimers; panel (**C**) E2-E1 dimers.

**Figure 9 jof-10-00795-f009:**
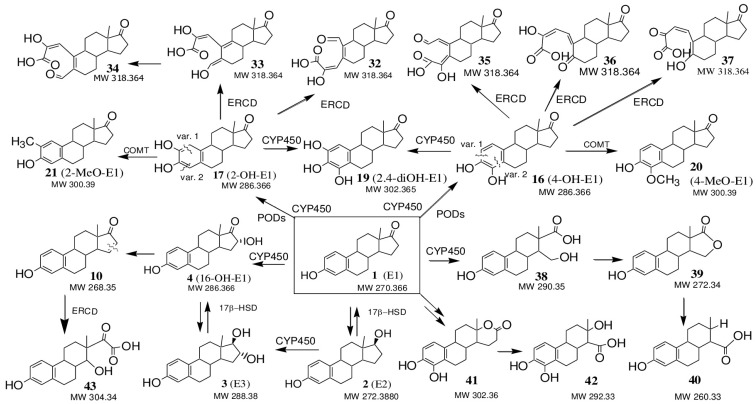
Proposed pathways for E1 transformation by the *T. hirsuta* LE-BIN 072 enzymes. ERCD- extradiol ring-cleavage dioxygenase, COMT- catechol O-methyltransferase, CYP450- cytochrome P-450-dependent monooxygenase, PODs- peroxidases, 17β-HSD- 17β-hydroxysteroid dehydrogenase.

**Figure 10 jof-10-00795-f010:**
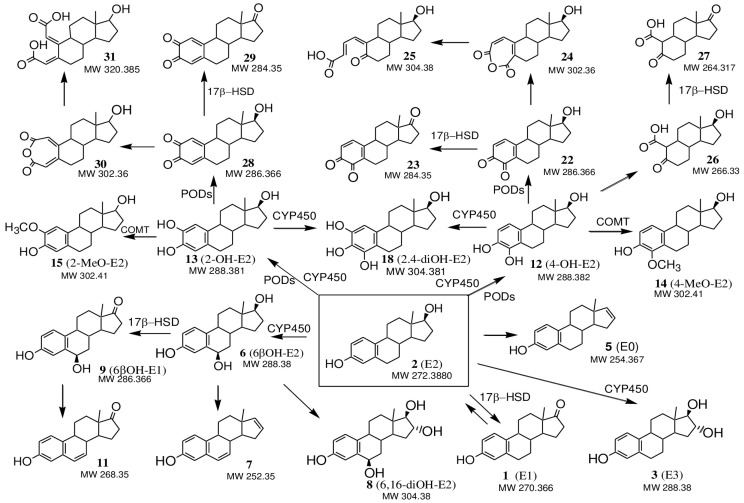
Proposed pathways for E2 transformation by the *T. hirsuta* LE-BIN 072 enzymes. ERCD- extradiol ring-cleavage dioxygenase, COMT- catechol O-methyltransferase, CYP450- cytochrome P-450-dependent monooxygenase, PODs- peroxidases, 17β-HSD- 17β-hydroxysteroid dehydrogenase.

**Table 1 jof-10-00795-t001:** The residual quantity of E1 and E2 extracted during the biotransformation process.

Cultivation Time, Days	Extracted E1, µg/mL			Extracted E2, µg/mL		
	CLs	Mycelium	Total *	CLs	Mycelium	Total *
1	2.7 ± 0.3	6.1 ± 0.6	8.8 ± 0.9	0.2 ± 0.02	7.2 ± 0.7	7.4 ± 0.1
3	2.3 ± 0.2	4.4 ± 0.4	6.7 ± 0.6	0.2 ± 0.01	0.6 ± 0.05	0.7 ± 0.06
6	1.2 ± 0.1	6.4 ± 0.6	7.6 ± 0.7	traces	0.1 ± 0.03	0.1 ± 0.03
10	1.9 ± 0.2	6.1 ± 0.6	8.0 ± 0.8	traces	traces	traces

*—initial concentration of oestrogens was 100 µg/mL.

**Table 2 jof-10-00795-t002:** List of proposed oestrone (E1) and 17β-oestradiol (E2) biotransformation products identified by HPLC-MS.

Products	Formula	Theoretical Molar Weight	Observed Rt (min)	Observed (Calculated) [M]^+^ *m*/*z*	Note	GP + E1 Extracts								GP + E2 Extracts							
						1 Day		3 Days		6 Days		10 Days		1 Day		3 Days		6 Days		10 Days	
						CL *	M *	CL	M	CL	M	CL	M	CL	M	CL	M	CL	M	CL	M
**1** (E1)	C_18_H_22_O_2_	270.3667	18.89	270.1623	obtained in this work	+	+	+	+	+	+	+	+	+	+	+	+	+	+	+	+
**2** (E2)	C_18_H_24_O_2_	272.3826	17.58	272.1778	obtained in this work	+	+	+	+	+	+	+	+	+	+	+	+	+	+	+	+
**3** (E3)	C_18_H_24_O_3_	288.3820	13.81	288.1717	obtained in this work	+	-	-	-	-	-	-	-	+	-	-	-	-	-	-	-
**4** (16-OH-E1) or **9** (6β-OH-E1)	C_18_H_22_O_3_	286.3661	15.35	286.156	287.31 [M + H]^+^ [50]	+	+	-	+	-	-	-	-	+	-	-	+	-	-	-	-
**5** (E0)	C_18_H_22_O	254.3673	18.23	254.1666	255.1749 [M + H]^+^ [51,52]	-	-	+	+	-	+	+	+	+	+	+	+	-	+	+	+
**6** (6β-OH-E2)	C_18_H_24_O_3_	288.3820	12.77	288.1731	presumably	+	-	-	-	-	-	-	+	+	-	-	+	-	-	-	+
**7**	C_18_H_20_O	252.3514	17.32	252.1348	presumably	-	-	+	-	-	-	-	-	-	-	+	-	+	-	+	-
**8**, **25**, **43**	C_18_H_24_O_4_	304.3814	15.74–23.9	304.1661	presumably	-	-	+	-	+	-	+	-	+	-	+	-	+	-	+	-
**10** or **11**	C_18_H_20_O_2_	268.3508	18.73	268.1458	267 [M-H]^−^ [53]	+	+	+	+	+	+	+	+	+	+	+	+	+	+	+	+
**12** (4-OH-E2) or **13** (2-OH-E2)	C_18_H_24_O_3_	288.3820	20.46	288.1133	289.1804 [M + H]^+^ [48,51]	+	-	-	-	+	-	+	-	+	+	-	-	+	-	+	-
**14** (4OMeE2) or **15** (2OMeE2)	C_19_H_26_O_3_	302.4086	19.3	302.1889	303.1960 [M + H]^+^ [48,51]	+	-	-	-	-	-	-	-	+	-	+	-	+	-	+	-
**16** (4-OH-E1), **17** (2-OH-E1), **22** or **28**	C_18_H_22_O_3_	286.3661	14.2–20.9	286.1588	287.27 [M + H]^+^ [50] 287.1648 [M + H]^+^ [52] 287.1647 [M + H]^+^ [51]	+	+	+	+	+	+	+	+	+	+	+	+	+	+	+	+
**18** (2,4-diOH-E2)	C_18_H_24_O_4_	304.3814	9.78	304.1678	presumably	+	-	+	-	+	-	+	-	-	-	+	-	+	-	+	-
**19** (2,4-diOH-E1)	C_18_H_22_O_4_	302.3655	13.06	302.1539	presumably	+	+	+	+	-	+	+	+	+	-	-	+	-	-	-	-
**20** (4-OMe-E1) or **21** (2-OMe-E1)	C_19_H_24_O_3_	300.3927	26.66	300.2084	presumably	+	+	+	+	+	+	+	+	+	+	+	+	+	+	+	+
**23** or **29**	C_18_H_20_O_3_	284.3502	20.44	284.166	presumably	-	-	-	+	-	+	-	+	-	-	-	+	-	+	-	+
**24**, **30**, **41**	C_18_H_22_O_4_	302.3655	30.11–30.8	302.283	301.1440 [M-H]^+^ [54] 303.1596 [M + H]^+^ [51]	+	+	+	+	+	+	+	+	+	+	+	+	+	+	+	+
**26**	C_15_H_22_O_4_	266.3333	14.84	266.1523	267.1596 [M + H]^+^ [51]	+	+	+	+	+	+	+	+	+	+	+	+	+	+	+	+
**27**	C_15_H_20_O_4_	264.3174	15.62	264.1731	presumably	+	+	+	+	+	+	+	+	+	+	+	+	+	-	+	-
**31**	C_18_H_24_O_5_	320.3808	9.11	320.1416	presumably	-	-	-	+	-	+	-	+	-	-	-	+	-	+	-	+
**32**–**37**	C_18_H_22_O_5_	318.3649	34.8–34.91	318.253	301.15 [M-H2O + H]^+^ [52,55]	+	+	+	+	+	+	+	+	+	+	+	+	+	+	+	+
**38**	C_17_H_22_O_4_	290.3548	13.17	290.1883	[56]	+	+	+	+	+	+	+	+	+	+	+	+	+	+	+	+
**39**	C_17_H_20_O_3_	272.3395	30.77	272.251	[56]	+	+	+	+	+	+	+	+	+	+	+	+	+	+	+	+
**40**	C_16_H_20_O_3_	260.3288	10.58	260.1247	[56]	+	+	-	+	-	-	-	-	+	+	-	+	-	-	-	-
**42**	C_16_H_20_O_5_	292.3276	25.07	292.1672	293.1389 [M + H]^+^ [51]	-	-	+	+	+	+	+	+	+	+	+	+	+	-	+	-
**44**, E1-E1 dimers	C_36_H_42_O_4_	538.7176	23.64	538.3064	presumably	+	+	+	+	+	+	+	+	+	+	+	+	+	+	+	+
**45**, E2-E1 dimers	C_36_H_44_O_4_	540.7335	18.89	540.324	presumably	+	+	+	+	+	+	+	+	+	-	-	+	-	+	+	+
**46**, E2-E2 dimers	C_36_H_46_O_4_	542.7493	22.4	542.341	541.3321 [M-H]^+^ [31]	-	-	+	-	+	+	+	+	+	+	+	+	+	+	+	+
**47**, (2-OH-E1)-E1, (4-OH-E1)-E1, dimeric quinone-like derivatives and other combinations	C_36_H_42_O_5_	554.7170	12.62	554.263	[31]	-	-	+	-	+	+	+	+	-	-	+	+	+	+	+	+
**48**, (2-OH-E2)-E2, (4-OH-E2)-E2, quinone-like derivatives and other combinations	C_36_H_46_O_5_	558.7487	21.43	558.3358	[31]	-	-	+	+	+	+	+	-	+	+	+	+	+	+	+	+
**49**, dimer without designation of the structure	C_36_H_34_O_7_	578.6523	25.67	578.2857	[31]	+	+	+	+	+	+	+	+	+	+	+	+	+	+	+	+

* CL—culture liquid extract; M—mycelium extract.

## Data Availability

The data are presented in the Appendix A to the article.

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
