# Peer review of "Oestrogen Detoxification Ability of White Rot Fungus *Trametes hirsuta* LE-BIN 072: Exoproteome and Transformation Product Profiling"

_jof, 2024, doi:10.3390/jof10110795_

Round 1

Reviewer 1 Report

In this work an in depth analysis of the proteomic events, along with analysis of the chemical nature of small molecules in the supernatant, during estrogen degradation by T. hirsuta is performed. Although many reports exist about the ability of white-rot fungi and their redox enzymes (mainly extracellular enzymes) to degrade estrogens, this work aims to find out what is going on at the molecular level, by examining both the extracellular proteins present at various stages of the culture, as well as the presence of degradation products. I think this work is a tour de force for the topic, and I recommend its publication. It will be very interesting for the readers, not only because it partially explains the observed degradation, but also because it opens new questions, for example the absence of laccases, which are usually present in this processes, and also about the role of other, less studied oxidases.

Title:  Metabolomics is not the right term for what the authors measured. As I understand they analyzed the supernatant of the cultures, and in some cases they extracted what was attached to the mycelium. They did not analyzed the intracellular material. So I would recommend using another term in the title and along the manuscript. Extracellular metabolomics may be more accurate

Figure 2. The initial phrase "Dynamics of T. hirsuta LE-BIN 072 biomass accumulation in..."" is a strange way to say "Growth curve of T. hisuta LE-BIN 072 in the GP medium....." etc. I recommend rephrasing that initial idea

Figure 3. The letters a, b,c d are missing in each panel

Figure 5. The phrase "Total dynamics...." is also an uncommon way to say "Degradation kinetics of estrone (E1) and estradiol (E2) by white rot...."etc. I recommend reconsidering that use of language

Figure 6. Similar comment, for the word dynamics used s synonym of kinetics. Instead of Dynamics of total (in culture liquid and mycelium extracts) E2 and E1 accumulation as biotransformation products;,I suggest "E2 and E1 accumulation as biotransformation products (in culture supernatant and mycelium extracts)..."etc

Table 2. Nowhere in the table I found out information on the abundance of the mentioned compounds. In the text, it is stated that major products are dimer, trimers. But info could not be found. I suggest including in the supplementary material an image of the HPLC chromatogram, highlighting the peaks corresponding to the major products. Otherwise the information is hard to analyze. There are so many metabolites, and for that reason it is important to mention its relative abundance

Line 513-514. The phrase "This may indicate that E1 (or its degradation products) are more preferred substrates for MnPs, while E2 (or its degradation products) are more preferred substrates for VP2." should be reconsidered. The enzymes are usually produced because the compounds acts as inducers of the expression at DNA level. Not the other way, as the text suggests.   

Author Response

Reviewer #1:

Comments: In this work an in depth analysis of the proteomic events, along with analysis of the chemical nature of small molecules in the supernatant, during estrogen degradation by T. hirsuta is performed. Although many reports exist about the ability of white-rot fungi and their redox enzymes (mainly extracellular enzymes) to degrade estrogens, this work aims to find out what is going on at the molecular level, by examining both the extracellular proteins present at various stages of the culture, as well as the presence of degradation products. I think this work is a tour de force for the topic, and I recommend its publication. It will be very interesting for the readers, not only because it partially explains the observed degradation, but also because it opens new questions, for example the absence of laccases, which are usually present in this processes, and also about the role of other, less studied oxidases.

Response: The authors thank the reviewer for the overall positive assessment of the manuscript and its careful consideration, which allowed us to significantly improve its quality.

Point by point answers are given below:

Comments: Title: Metabolomics is not the right term for what the authors measured. As I understand they analyzed the supernatant of the cultures, and in some cases they extracted what was attached to the mycelium. They did not analyzed the intracellular material. So I would recommend using another term in the title and along the manuscript. Extracellular metabolomics may be more accurate

Response: Corrected. The evaluation of estrogen transformation products was carried out both in the culture liquid and in the fungal mycelium extracts (see the methodology, paragraph 2.5). In the title and further in the text the term "metabolome" was replaced by "transformation products profiling". We have changed the title to: “Estrogen detoxification ability of white rot fungus Trametes hirsuta LE-BIN 072: exoproteomes and transformation products profiling”.

Comments: Figure 2. The initial phrase "Dynamics of T. hirsuta LE-BIN 072 biomass accumulation in..." " is a strange way to say "Growth curve of T. hisuta LE-BIN 072 in the GP medium....." etc. I recommend rephrasing that initial idea

Response: Corrected to: “Figure 2. Growth curves of T. hisuta LE-BIN 072 in the GP medium (black line), GP + 5% methanol (gray line, GP+Met), GP+E1 in methanol (yellow line) and GP+E2 in methanol (red line).

Comments: Figure 3. The letters a, b, c d are missing in each panel

Response: Corrected

Comments: Figure 5. The phrase "Total dynamics...." is also an uncommon way to say "Degradation kinetics of estrone (E1) and estradiol (E2) by white rot...."etc. I recommend reconsidering that use of language

Response: Following the recommendations of other reviewer, we have removed Figure 5 from the manuscript. Now these experimental data are presented as Table 1. The term “Total” in this table reflects sums of concentrations for CLs and mycelium. Data for different time of the cultivation are presented as different rows of the table and are not specified as kinetics or dynamics. Discussion of the Table 1 has been additionally checked for the absence of the «dynamics» term.

Comments: Figure 6. Similar comment, for the word dynamics used s synonym of kinetics. Instead of Dynamics of total (in culture liquid and mycelium extracts) E2 and E1 accumulation as biotransformation products; I suggest "E2 and E1 accumulation as biotransformation products (in culture supernatant and mycelium extracts)..."etc

Response: Corrected. Figure 6 in the new version is designated as Figure 5. E2 and E1 accumulation as biotransformation products (in culture supernatant and mycelium extracts); (a) - E1 is initial substrate (GP+E1 medium); (b) - E2 is initial substrate (GP+E2 medium).

Comments: Table 2. Nowhere in the table I found out information on the abundance of the mentioned compounds. In the text, it is stated that major products are dimer, trimers. But info could not be found. I suggest including in the supplementary material an image of the HPLC chromatogram, highlighting the peaks corresponding to the major products. Otherwise the information is hard to analyze. There are so many metabolites, and for that reason it is important to mention its relative abundance.

Response: Figures S2 and S3 (Figure S2: HPLC-MS chromatograms (TIC) of culture liquid (CL) and mycelium extracts Trametes hirsuta LE-BIN 072 during cultivation on glucose-peptone (GP) medium supplemented with E1 (GP+E1). The designations of the compounds are presented in Table 1; Figure S3: HPLC-MS chromatograms (TIC) of culture liquid (CL) and mycelium extracts Trametes hirsuta LE-BIN 072 during cultivation on glucose-peptone (GP) medium supplemented with E2 (GP+E2). The designations of the compounds are presented in Table 1.) have been added to the suppl. material, and peaks corresponding to the main products are labeled. Additional description has been added to the text (lines 380-389).

Comments: Line 513-514. The phrase "This may indicate that E1 (or its degradation products) are more preferred substrates for MnPs, while E2 (or its degradation products) are more preferred substrates for VP2." should be reconsidered. The enzymes are usually produced because the compounds acts as inducers of the expression at DNA level. Not the other way, as the text suggests.  

Response: Corrected to:” This may indicate that E1 (or its degradation products) are likely to induce MnP expression to a greater extent, whereas E2 (or its degradation products) are likely to induce VP2 expression.” (lines 547-550).

Reviewer 2 Report

This work is well organized, and the quality of this manuscript should be further improved. Major revision is suggested.

1. In Figure 3a (10 day), Figure 3b (6 day), and Figure 3c (6 day), why the enzymatic activity decreased significantly and even declined to zero? The potential reason need be clarified and related references should be cited.

2. Figure 5 is hard to understand. The differences of E1 and E2 are not significant.

3. Significant Digit in Table 1 are not appropriate.

4. Based on the Table 2, Figure 7 and Figure 8, potential mechanism and main products and by-products should be briefly summarized in one scheme in order to make readers well understand this work. The information about Figure S2, S3, S4, S5, S6 and S7 need be further shortened.

5. The Captions of Figure S1 need be provided in details.

Author Response

Reviewer #2:

Comments: This work is well organized, and the quality of this manuscript should be further improved. Major revision is suggested.

Response: We are grateful to reviewer for valuable comments and recommendations. This allowed us to significantly improve the manuscript. Point by point answers are given below:

Comments: 1. In Figure 3a (10 day), Figure 3b (6 day), and Figure 3c (6 day), why the enzymatic activity decreased significantly and even declined to zero? The potential reason need be clarified and related references should be cited.

Response: Corrected. A technical error occurred while preparing the graph Figure 3a, corrections have been made. The necessary explanation regarding the curves in Figure 3b (6 day), and Figure 3c (6 day), as well as the reference, have been added to the text (lines 283-289). Corrected to: “Similar oxidative enzyme activity profiles in the fungal CLs with the peaks and subsequent decrease to trace values were previously shown during the cultivation of T. hirsuta LE-BIN 072 on media such as GP in the presence of CuSO4 and GP supplemented with lignocellulose substrate – milled oat straw [45]. As for esterase activity, a noticeable increase in this activity was also observed on the 3rd day of cultivation in the presence of estrogens, in contrast to GP and GP+Met (Figure 3d).”

Trace values of oxidative enzyme activities on the 6th day of fungal cultivation are also discussed in the sections “Results” (lines 492-494); and “Discussion” (lines 531-538) and are associated with the absence of oxidative enzymes in fungal secretomes. Corrected to: “The results of the present study showed that by the 3rd day of culturing T. hirsuta LE-BIN 072 in the presence of estrogens, the total residual amount of the initial E1 and E2 was less than 10% (Table 1). At this time point, the activity of oxidative enzymes was maximum (Figure 3) and 3 ligninolytic peroxidases were identified in the secretomes - MnP5, MnP7 and VP2. While on the 6th day, peroxidases were absent in the exoproteoms (Figure 6), which correlated with the absence of oxidative activity in the CLs (Figure 3 b and c).”

Comments: 2. Figure 5 is hard to understand. The differences of E1 and E2 are not significant.

Response: Figure 5 has been removed from the manuscript and the data are presented in tabular form (see Table 1, column “Total”) for extended and more focused comments to each obtained value. The corresponding changes were made to the text: “The experiment showed that after 24 h of cultivation, the total residual amount of both E1 and E2 was about 8% and did not change significantly during further cultivation for E1. For E2 traces amounts was detected starting from day 3 (Table 1, columns Total)” (lines 332-337).

Comments: 3. Significant Digit in Table 1 are not appropriate.

Response: Corrected in table 1

Comments: 4. Based on the Table 2, Figure 7 and Figure 8, potential mechanism and main products and by-products should be briefly summarized in one scheme in order to make readers well understand this work. The information about Figure S2, S3, S4, S5, S6 and S7 need be further shortened.

Response: We thank the reviewer for valuable comments. Proposed pathways for E1 and E2 biotransformation by the T. hirsuta are presented in the new schemes on Figures 8, 9 and 10 in the text. Respectively figures S2, S3, S4, S5, S6 and S7 have been removed from the Supplementary material. Due to significant quantity (> 40) of the compounds considered in these three schemes, we cannot integrate all of them to one scheme.

Comments: 5. The Captions of Figure S1 need be provided in details.

Response: Corrected to:” Two-dimensional gel electrophoresis (2DE) of the Trametes hirsuta LE-BIN 072 exoproteomes obtained during its cultivation on the control glucose-peptone (GP) medium with methanol (GP+Met) and GP medium supplemented with E1 (GP+E1) and with E2 (GP+E2). Proteins secreted in GP+Met, GP+E1 and GP+E2 media are shown on 1+2+3 day of cultivation (top panel), on 6th day of cultivation (central panel) and 10th day of cultivation (bottom panel). The designation “1+2+3 day” means that culture broth from days 1, 2, and 3 of cultivation were pooled together. For the data on MALDI TOF/TOF MS/MS analysis of the highlighted protein spots, please, refer to Figure 6.”

Round 2

Reviewer 2 Report

This revised version can be accepted as it is.

This revised version can be accepted as it is.